



**Temperature exerted no influence on the organic carbon**
**isotope of surface soil along the isopleth of 400 mm mean**
**annual precipitation in China**

5          Yufu Jia,    Guoan Wang, Qiqi Tan, and    Zixun Chen

[1]College of Resources and Environmental Sciences, China Agricultural University,
Beijing 100193, China
Author for correspondence:
Guoan Wang
Tel: +086-10-62733942
Email:gawang@cau.edu.cn





## Abstract

Soil organic carbon is the largest pool of terrestrial ecosystem and its carbon isotope composition is affected by many factors. However, the influence of environmental factors, especially temperature, on soil organic carbon isotope ($\delta^{13}C_{SOM}$) is poorly constrained. This impedes interpretations and application of variability of organic carbon isotope in reconstructions of paleoclimate and paleoecology and global carbon cycling. With a considerable temperature gradient along the 400 mm isohyet (isopleth of mean annual precipitation – MAP) in China, this isohyet provides ideal experimental sites for studying the influence of temperature on soil organic carbon isotope. In this study, the effect of temperature on surface soil $\delta^{13}C$ was assessed by a comprehensive investigation from 27 sites across a temperature gradient along the isohyet. This work demonstrates that temperature did not play a role in soil $\delta^{13}C$, this suggests that organic carbon isotopes in sediments cannot be used for the paleotemperature reconstruction, and that the effect of temperature on organic carbon isotopes can be neglected in the reconstruction of paleoclimate and paleovegetation. Multiple regression with MAT (mean annual temperature), MAP, altitude, latitude and longitude as independent variables, and $\delta^{13}C_{SOM}$ as the dependent variable, shows that the five environmental factors in total account for only 9% soil $\delta^{13}C$ variance. However, One-way ANOVA analyses suggest that soil and vegetation types are significant influential factors on soil $\delta^{13}C$. Multiple regressions in which above five environmental factors were taken as quantitative variables, vegetation type, Chinese nomenclature soil type and WRB soil type were introduced as dummy variables separately, show that 36.2%, 37.4%, 29.7% of the variability in soil $\delta^{13}C$ are explained, respectively. Compared to the multiple regression in which only quantitative environmental variables were introduced, the multiple regressions in which soil and vegetation were also introduced explain more variance, suggesting that soil type and vegetation type really exerted significant influences on $\delta^{13}C_{SOM}$.



## 1. Introduction

Global climate change has recently received a great deal of attention, and effective

predictions of future climate change depend on the relevant information from climate

in the geological past. Over recent decades, stable carbon isotopes in sediments, such

as loess, paleosol, lacustrine and marine sediments, have been widely used to

reconstruct paleovegetation and paleoenvironments, and provided important insights

into patterns of past climate and environment changes. For examples, many

researchers have used organic carbon isotopes of loess to reconstruct paleovegetation

and paleoprecipitation. Vidic and Montañez (2004) conducted a reconstruction of

paleovegetation at the central Chinese Loess Plateau during the Last Glaciation (LG)

and Holocene by using the organic carbon isotopes in loess from Jiaodao, Shanxi

Province. Hatté and Guiot (2005) carried out a palaeoprecipitation reconstruction by

inverse modelling using the organic carbon isotopic signal of the Nußloch loess

sequence (Rhine Valley, Germany). Rao et al. (2013) reconstructed a high-resolution

summer precipitation variations in the western Chinese Loess Plateau during the Last

Glaciation using a well-dated organic carbon isotopic dataset. Yang et al. (2015)

derived a minimum 300 km northwestward migration of the monsoon rain belt from

the Last Glacial Maximum to the Mid-Holocene using the organic carbon isotopes

from 21 loess sections across the Loess Plateau. However, to our knowledge, almost

no researchers have conducted paleotemperature reconstructions using organic carbon

isotope records of loess and paleosol, because it has been argued that temperature

exerts slight, or even no influence on $\delta^{13}C_{SOM}$. While this statement may be likely, it



needs to be demonstrated because only few studies have addressed the influence of
temperature on organic carbon isotopes of modern surface soil; furthermore, these
studies do not appear to result in a conclusive statement. Lee et al. (2005) and Feng et
al. (2008) both reported no relationship between temperature and surface soil $\delta^{13}$C in
central-east Asia. However, Lu et al. (2004) discovered a nonlinear relationship
between annual mean temperature (MAT) and $\delta^{13}C_{SOM}$ from the Qinghai-Tibetan
Plateau; Sage et al. (1999) compiled the data from Bird and Pousai (1997) and also
found a nonlinear trend for the variation in $\delta^{13}C_{SOM}$ along a temperature gradient in
Australian grasslands and savannas.
Plant residues are the most important source of soil organic matter. $\delta^{13}C_{SOM}$ is
generally close to plant carbon isotope despite isotopic fractionation during
decomposition of organic matter (Nadelhoffer and Fry, 1988; Balesdent et al., 1993;
Ågren et al., 1996; Fernandez et al., 2003; Wynn, 2007). Thus, the influential factors
of plants $\delta^{13}$C might also play a role in $\delta^{13}C_{SOM}$, $\delta^{13}$C in plants, especially $C_3$ plants, is
tightly associated with precipitation (Diefendorf et al., 2010; Kohn, 2010), so,
precipitation should have influence on soil $\delta^{13}$C. In addition to effect of precipitation,
many factors, such as temperature, air pressure, atmospheric $CO_2$  concentration,
altitude, latitude and longitude, may also exert influences on variance in plants $\delta^{13}$C
(Körner et al., 1991; Hultine and Marshall, 2000; Zhu et al., 2010; Xu et al., 2015).
Although patterns of variation in plants $\delta^{13}$C with temperature are unresolved so far
(e.g. Schleser et al., 1999; McCarroll and Loader, 2004; Treydte et al., 2007; Wang et
al., 2013), it has been widely accepted that, even if temperature has effect on plants





$\delta^{13}$C, this effect is slight. So, if the $^{13}$C enrichment during SOM decomposition is a
constant value, we expect a slight or no influence of temperature on soil $\delta^{13}$C.
However, the fact is that this $^{13}$C-enrichment is affected by environmental and biotic
factors (Wang et al., 2015). Thus, it is difficult to expect whether or how temperature
affects soil $\delta^{13}$C, and it needs specific investigations of focusing on this issue.
Although the relationship between temperature and $\delta^{13}$C$_{SOM}$ has been investigated in
these previous studies mentioned above, these studies were unable to effectively
separate the influence of temperature from the effect of precipitation. Thus, new
investigations are necessary. The present study includes an intensive investigation of
the variation in $\delta^{13}$C$_{SOM}$ with temperature across a temperature gradient along the 400
mm isohyet (isopleth of mean annual precipitation - MAP) in China. We sampled
surface soil along the specific isohyet to minimize the effect of precipitation changes
on $\delta^{13}$C$_{SOM}$.
In addition, there are no meteorological stations near most of the sampling sites in
the previous studies mentioned above; thus, they had to interpolate meteorological
data, which could be unrealistic in regions with strong topographical variability. This
interpolation could produce errors in the relationships between temperature and
$\delta^{13}$C$_{SOM}$ established in these studies. In the present investigation, we collected samples
only at those sites with meteorological stations; thus, the climatic data that we
obtained from these stations are probably more reliable compared with the
pseudo-data derived by interpolation.





## 2. Materials and methods


2.1. Study site
In this study, we set up a transect along the 400 mm isohyet from LangKaZi (site 1,
29°3.309′N, 90°23.469′E), on the Qinghai-Tibetan Plateau in southwest China, to
BeiJiCun (Site27, 53°17.458′N, 122°8.752′E), in Heilongjiang Province in northeast
China (Fig.1, Table 1). The straight-line distance between the above two sites is about
6000 km. Twenty-seven (27) sampling sites were set along the transect. Among these
sampling sites, 10 sites are located on the Qinghai-Tibetan Plateau, and the others are
in north China. BeiJiCun and KuDuEr have the lowest MAT of -5.5 $^{\circ}$C and ShenMu
has the highest MAT of +8.9 $^{\circ}$C. The average MAP of these sites is 402 mm. In north
China, rainfall from June to September accounts for approximately 80% of the total
annual precipitation, and the dominant control over the amount of precipitation is the
strength of the East-Asian monsoon system. In the Qinghai-Tibetan Plateau, however,
precipitation is associated with both the Southwest monsoon and the Qinghai-Tibetan
Plateau monsoon, and approximately 80% - 90% rainfall occurs in the summer season
(from May to October).

135        Fig.1

136        Table 1

2.2 soil sampling
Soil samples were collected in the summer of 2013 between July 12th and August
30th. In order to avoid disturbance of human activities, sample sites are 5-7



kilometers far from the towns where the meteorological stations are located. We set
three quadrats (0.5 m×0.5 m) within 200 $m^2$ to collect surface mineral soil (0-5 cm)
using a ring knife. The O-horizon, including litters, moders and mors were removed
before collecting mineral soil. About 10 g air dried soils were sieved at 2 mm. Plant
fragments and the soil fraction coarser than 2 mm were removed. The rest of the soil
samples were immersed using excessive HCl (1 mol/L) for 24 h. In order to ensure
that all carbonate was cleared, we conducted artificial stirring 4 times during the
immersion. Then, the sample was washed to neutrality using distilled water. Finally it
was oven-dried at 50℃ and ground. Carbon isotope ratios were determined on a
Delta$^{Plus}$XP mass spectrometer (Thermo Scientific, Bremen, Germany) coupled with
an elemental analyzer (FlashEA 1112; CE Instruments,Wigan, UK) in continuous
flow mode. The elemental analyzer combustion temperature was 1020 $^o$C.
The carbon isotopic ratios are reported in delta notation relative to the V-PDB
standard using the equation:
$$\delta^{13}C = (R_{sample}/R_{standard} - 1) \times 1000 \qquad (1)$$
where $\delta^{13}C$ is the carbon isotope ratio of the sample (‰), and $R_{sample}$ and $R_{standard}$ are
the $^{13}C/^{12}C$ ratios of the sample and the standard, respectively. For this measurement,
we obtained a standard deviation of less than 0.15‰ among replicate measurements
of the same soil sample.

## 3. Results


Except for one $\delta^{13}C_{SOM}$ value (-18.8‰), all other data range from -20.4‰ to -27.1‰





with a mean value of -23.3‰ (n =80, s.d. =1.45). Multiple regression with MAT,
MAP, altitude, latitude and longitude as independent variables, and $\delta^{13}C_{SOM}$ as the
dependent variable, shows that only 9% of the variability in soil $\delta^{13}C$ can be explained
as a linear combination of all five environmental factors (p = 0.205) (Table 2).
Considering the possibility of correlations among the five explanatory variables,
stepwise regression was used to eliminate the potential influence of collinearity
among them. Variables were incorporated into the model with $P$-value < 0.05 and
exclude with $P$-value > 0.1. Stepwise regression of soil $\delta^{13}C$ in the model consisting
only of latitude ($R^2$ = 0.077, p = 0.012). In order to constrain the relationship between
soil $\delta^{13}C$ and each environmental factor better, bivariate correlation analyses of soil
$\delta^{13}C$ against some environmental factors were conducted. The bivariate correlation
analyses show that $\delta^{13}C_{SOM}$ is not related to MAT (p = 0.114) or SMT (p = 0.697)
along the isohyet (Fig. 2a, b). In addition, in order to determine further the response of
$\delta^{13}C_{SOM}$ to temperature, we considered three subsets of our soil samples defined
according to the climate, topography or vegetation type: the Qinghai–Tibetan Plateau
(mainly alpine meadow, including 10 sites), steppe or grassland (11 sites) and
coniferous forest (6 sites) (Table 1). Bivariate correlation analyses within these
subsets also show no relationship between $\delta^{13}C_{SOM}$ and MAT for all categories. The
correlation analysis of $\delta^{13}C_{SOM}$ vs. altitude is shown in Fig.3, which displays no
relationship (p = 0.132). Although longitude is not found to exert influence on
$\delta^{13}C_{SOM}$ in the above stepwise regression, bivariate correlation analyses show that
latitude and longitude both are negatively related to $\delta^{13}C_{SOM}$ (p =0.012 and 0.034,





184 respectively) (Fig. 4a,b).

185  In addition to effects of quantifiable environmental factors,qualitative factors, such

186 as soil type and vegetation type, may have influence on $\delta^{13}C_{SOM}$. Varied concepts

187 have been introduced in soil taxonomy, leaving varied soil nomenclatures in use. In

188 this study we adopted Chinese soil nomenclature and the World Reference Base

189 (WRB) to describe the observed soil. The soil was divided into 8 types and 6 types

190 based on the Chinese Soil Taxonomy and WRB, respectively (Table 1). One-way

191 ANOVA analyses suggest that soil type and vegetation type both played a significant

192 role in $\delta^{13}C_{SOM}$ (p = 0.002 for soil types based on the Chinese Soil Taxonomy, p =

193 0.003 for soil type based on WRB and p = 0.001 for vegetation types) (Fig. 5).

194  In order to constrain further the effects of soil type and vegetation type on $\delta^{13}C_{SOM}$,

195 multiple regressions with soil type and vegetation type as dummy variables were

196 conducted. Considering the tight relationship between soil type and vegetation type,

197 especially in Chinese soil taxonomy, the soil variable and the vegetation variable were

198 separately introduced into the statistical analyses. Multiple regression, in which the

199 above five explanatory environmental factors were taken as quantitative variables and

200 the 8 soil types of the Chinese nomenclature as values of a dummy variable, shows

201 that environmental factors and soil types in total account for 37.4% soil $\delta^{13}C$ variance

202 (p < 0.001) (Table 2). 29.7% (p = 0.003) of the variability is explained using the 6 soil

203 types based on WRB rather than the Chinese nomenclature (Table 2). Similarly,

204 multiple regression with vegetation types as dummy variables shows that the five

205 environmental factors and vegetation types in total can explain 36.2% of the



variability in soil $\delta^{13}C$ (p = 0.001) (Table 2). Compared to the multiple regressions in
which only quantitative environmental variables were introduced, the multiple
regressions in which soil and vegetation were also introduced explain more variance,
suggesting that soil type and vegetation type really played a significant role in
$\delta^{13}C_{SOM}$ variability.
Table 2
Fig.2a, b
Fig.3
Fig.4a, b
Fig.5

## 4. Discussion

Soil $\delta^{13}C$ depends on plants $\delta^{13}C$ and carbon isotopic fractionation during organic
matter decomposition. $\delta^{13}C$ values of $C_3$ plants vary between −22‰ and −34‰ with a
mean of −27‰, and $C_4$ plants range from −9‰ to −19‰ with a mean of −13‰
(Dienes,1980). Carbon isotope fractionation occurs in the process of plant litter
decomposition into soil organic matter in most environments, especially in non-arid
environments, causing $^{13}C$-enrichment in soil organic matter compared with the plant
sources (Nadelhoffer, 1988; Balesdent et al., 1993; Ågren et al., 1996; Fernandez et
al., 2003; Wynn et al., 2005; Wynn, 2007). An intensive investigation of isotope
fractionation during organic matter decomposition, which was conducted in Mount
Gongga, an area in the Qinghai-Tibetan Plateau dominated by $C_3$ vegetation with



herbs, shrubs and trees, showed that the mean $^{13}$C-enrichment in surface soil (0-5 cm
depth ) relative to the vegetation was 2.87‰ (Chen et al., 2009). Another
investigation of 13 soil profiles from the Tibetan Plateau and north China showed the
$\delta^{13}$C difference between surface soil (0-5 cm depth ) and the original biomass varied
from 0.6 to 3.5‰ with a mean of 1.8‰ (Wang et al., 2008). Thus, the $\delta^{13}$C$_{SOM}$ data set
of this study ($\delta^{13}$C$_{SOM}$ ranges from -20.4‰ to -27.1‰) indicates that the modern
terrestrial ecosystem along the isohyet is greatly dominated by C$_3$ plants. This result is
consistent with the observations of vegetation along the isohyet done in our previous
study (Wang et al., 2013) and in this present study. Yin and Li (1997), Lu et al. (2004)
and Wang et al. (2004) have reported that a small number of C$_4$ species occurred in
the Qinghai-Tibetan Plateau; however, in this present study we found no C$_4$ plants in
the Qinghai-Tibetan Plateau. We are also very surprised at such high soil $\delta^{13}$C values
at RiKaZe (site 2) (Fig.3 and Table 1) because only four C$_3$ plants grow there, no C$_4$
species. The abnormal observation suggests that a very high carbon isotope
fractionation with SOM degradation have taken place in the local ecosystem.
Although the mechanism accounted for the unusually high isotopic fractionation
remains unclear, it is not surprising. For example, Wynn (2007) has reported that the
fractionation leaved soil organic carbon $^{13}$C-enriched by up to ∼6‰ with respect to
the original biomass. Rao et al. (2008) has suggested that mid-latitude area
(31°N-40°N) in east China provides relatively favorable condition for C$_4$ plant growth.
But we observed that a small number of C$_4$ species occur only in the temperate
meadow steppe and the temperate typical steppe in north China, while no C$_4$ species



are distributed in the coniferous forests in north China. In short, the contribution of $C_4$
biomass to the local vegetation along the isohyet is very low, and can be neglected.

252        The MAT, MAP, altitude, latitude and longitude, combined, are responsible for

only 9% variability in soil $\delta^{13}C$ in the multiple regression model, suggesting that the
contribution of the five environmental factors to the soil $\delta^{13}C$ variance is very small.
Our previous study conducted along the isohyet resulted in a strong positive
relationship between $C_3$ plant $\delta^{13}C$ and MAT with a coefficient of 0.104‰/$^{o}C$ (Wang
et al., 2013). The difference between maximum and minimum temperature along the
isohyet is 15$^{o}C$, so the greatest possible effect of temperature on plant $\delta^{13}C$ along the
temperature gradient is 1.56‰, which is not very great. Since the main source of soil
organic matter along the isohyet is $C_3$ plants, the induced variance in soil $\delta^{13}C$ by
plant $\delta^{13}C$ also cannot be very great. On the other hand, although the $^{13}C$-enrichment
with SOM degradation follows a Rayleigh distillation process (Wynn, 2007), our
recent study shows that temperature does not influence carbon isotopic fractionation
during decomposition (Wang et al., 2015), which is also a reason for the lack of a
relationship between soil $\delta^{13}C$ and temperature. Feng et al. (2008) and Lee et al. (2005)
respectively, reported no relationships between soil $\delta^{13}C$ and MAT and SMT, which
is consistent with our result. Their field campaigns were conducted in central Asia,
which is also dominated by $C_3$ plants, similar to the area along the 400 mm isohyet.
This is the reason why the same pattern exists in central Asia and the area along the
400 mm isohyet.

271        The observations in Bird and Pousai (1997) and Sage et al. (1999) appear to be



inconsistent with our findings; they found a nonlinear relationship between soil $\delta^{13}C$
and MAT in Australian grasslands. However, if considering only soil with pure $C_3$
plants (MAT is below 16℃), soil $\delta^{13}C$ and temperature are not related in Australian
grasslands, which is in agreement with our result. Below15℃, $C_4$ contribution to
productivity in Australian grasslands is negligible, whereas above 23 ℃ , $C_3$
contribution is negligible; Between 14℃ and 23℃,soil $\delta^{13}C$ is positively correlated
with MAT, indicating $C_4$ representation increasing with MAT (Sage et al., 1999). Lu
et al. (2004) also reported a nonlinear relationship between soil $\delta^{13}C$ and MAT.
Similarly, if the soil data with $C_4$ plants are excluded from the nonlinear correlation,
soil $\delta^{13}C$ is also not related to MAT in Lu et al. (2004) (see Fig.5b in Lu et al., 2004).
Thus, this present study and the previous observations are consistent in showing that
in a terrestrial ecosystem in which the vegetation is dominated by $C_3$ plants,
temperature does not influence soil $\delta^{13}C$ variance.
This study shows that the contribution of precipitation to the variability in soil $\delta^{13}C$
is neglected. The reason for this is that the soil was sampled along the 400 mm
isohyet, and the MAP difference among sites is very small. It should be pointed out
here that the no MAP influence on the soil $\delta^{13}C$ does not mean no moisture control of
the soil $\delta^{13}C$. Because the temperature varies greatly across the temperature gradient
although the MAP is almost the same for each sampling site ; this would cause a big
difference in relative humidity among sites. We expect that relative humidity would
explain a great variability in soil $\delta^{13}C$. But we did not take relative humidity as an
explanatory variable in the statistical analyses, because we lack the complete data of



relative humidity, and we do not want to use the pseudo-data derived by
interpolation.
Although stepwise regression and correlation analysis both show a significant
influence of latitude on soil $\delta^{13}$C, the five environmental variables including latitude
were responsible for only 9% variability in soil $\delta^{13}$C in a multiple regression model
(Table 2), suggesting that the contribution of latitude to soil $\delta^{13}$C was also slight. This
study shows a negative correlation between latitude and $\delta^{13}$C$_{SOM}$ (p=0.012). Bird and
Pausai (1997) and Tieszen et al. (1979) reported a similar pattern. Latitude is a
comprehensive environmental factor, and change in latitude can bring about changes
in other environmental factors, such as temperature, irradiation, cloud amount, and
moisture, but temperature or irradiation should be most strongly related to latitude,
and obviously change with latitude. The observed significant relationship between
latitude and soil $\delta^{13}$C (Fig.4a) suggests that environmental factors other than
temperature might contribute more or less to the variance in soil $\delta^{13}$C.
Vegetation type control of the soil $\delta^{13}$C mainly reflected the effects of life-form on
plant $\delta^{13}$C and substrate quality on isotope fractionation during organic matter
decomposition. Communities in which life-form of the dominant plants is similar are
generally treated as the same vegetation type. Plant $\delta^{13}$C is tightly related to life-form
(Diefendorf et al., 2010; Ehleringer and Cooper, 1988) and this causes $\delta^{13}$C
differences among varying vegetation types, consequently resulting in the observed
effect of vegetation type on the soil $\delta^{13}$C.
Substrate quality partly quantifies how easily organic carbon is used by soil





microbes (Poage and Feng, 2004). It can be related to plant type and is often defined
using a C/N ratio, lignin content, cellulose content, and/or lignin content/N ratio
(Melillo et al., 1989; Gartern et al., 2000). Our study in Mount Gongga, China,
showed that litter quality play a significant role in isotope fractionation during organic
matter decomposition, and the carbon isotope fractionation factor, α, increases with
litter quality (Wang et al., 2015). Thus, the isotope fractionation factor should be
different among varying sites because litter quality is dependent on vegetation and
this makes soil change its $\delta^{13}C$ with vegetation type.
Control of soil type on soil $\delta^{13}C$ could be associated with the effect of soil type on
isotope fractionation during organic matter decomposition, and involve at least two
mechanisms. (1) Properties and compositions of microbial decomposer communities
are dependent on soil type (Gelsomino et al., 1999). Different microbes could have
different metabolic pathways even when they decompose the same organic compound
(Macko   and   Estep,   1984),   and   the   extent   of   isotope   fractionation   during
decomposition may be tightly related to the metabolic pathways of microbes (Macko
and Estep, 1984). For example, Morasch et al. (2001) observed a greater hydrogen
isotope fractionation for toluene degradation in growth experiments with the aerobic
bacterium *P. putida* mt-2 and less fractionation in toluene degradation by anaerobic
bacteria. (2) Physical and chemical properties, such as pH, particle size fraction,
water-holding capacity, display striking differences among soil types and this causes
organic compounds to be decayed at different rate in different soil environments. The
magnitude of isotope fractionation during decomposition is linked to degree of





organic matter decomposition (Feng, 2002), thus, soil type plays a significant role in
fractionation.

**5. Conclusions**
The present study measured organic carbon isotopes in surface soil along a 400 mm
isohyet of mean annual precipitation in China, and observed that soil type and
vegetation type both had significant influence on soil organic carbon isotopes.
However, temperature is found to have no observable impact on $\delta^{13}C_{SOM}$, suggesting
that $\delta^{13}C$ signals in sediments cannot be used for the reconstruction of temperature,
and that the effect of temperature on $\delta^{13}C_{SOM}$ could be neglected in the reconstruction
of paleoclimate and paleovegetation using carbon isotopes of soil organic matter.


**Acknowledgments**
This research was supported by grants from the National Basic Research Program
(2014CB954202), the National Natural Science Foundation of China (No. 41272193)
and the China Scholarship Council (File No.201506355021). We would like to thank
Ma Yan for analyzing stable carbon isotope ratios in the Isotope Lab at the College of
Resources and Environment, China Agricultural University; we would also like to
thank Professor Eric S. Posmentier in the Department of Earth Sciences of Dartmouth
College for his constructive suggestions and English editing for this manuscript.





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



Figures


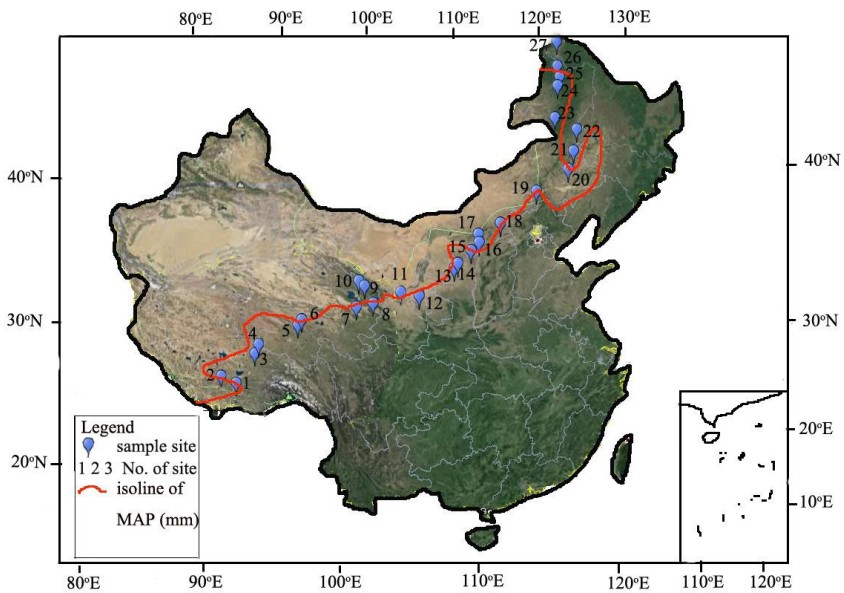



Fig.1. Sketch of sampled region. Sample sites are indicated with numbers. 1, LangKaZi; 2,
RiKaZe; 3, NaQu; 4, NieRong; 5, ZhiDuo; 6, QuMaLai; 7, TongDe; 8, TongRen; 9, HuangYuan;
10, HaiYan; 11, YuZhong; 12, XiJi; 13, JingBian; 14, HengShan; 15, ShenMu; 16, HeQu; 17,
ZhunGeErQi; 18, FengZhen; 19, DuoLun; 20, LinXi; 21, ZhaLuTeQi; 22, WuLanHaoTe; 23,
AErShan; 24, YaKeShi; 25, KuDuEr; 26, GenHe; 27, BeiJiCun. Detailed information of sites is
shown in Table 1.

















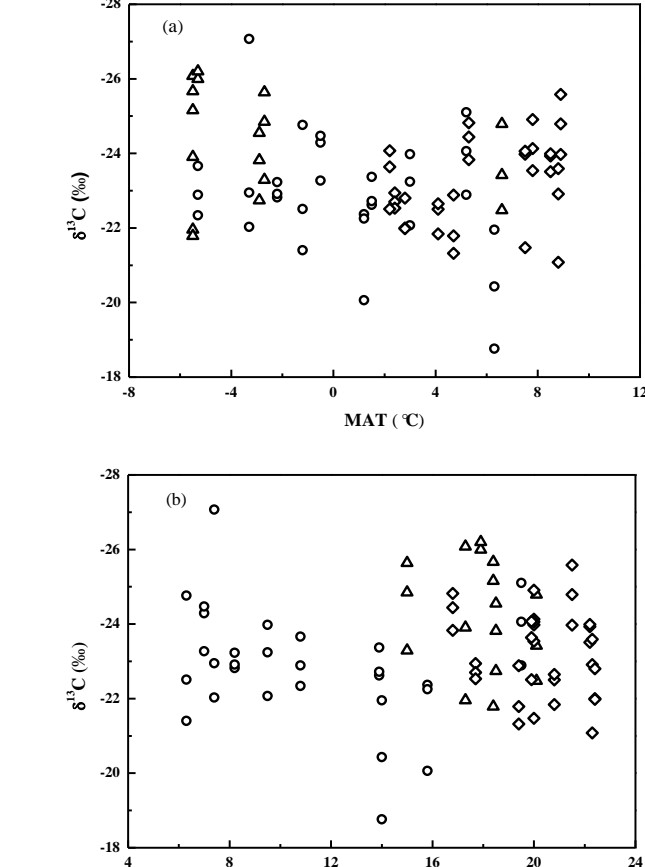




Fig.2 shows the variance in surface soil $\delta^{13}C$ with MAT (a) and SMT (b) along the 400 mm isoline
in China. Circle represents alpine and subalpine; diamond indicates temperate steppe and
grassland;, triangle is coniferous forest.
















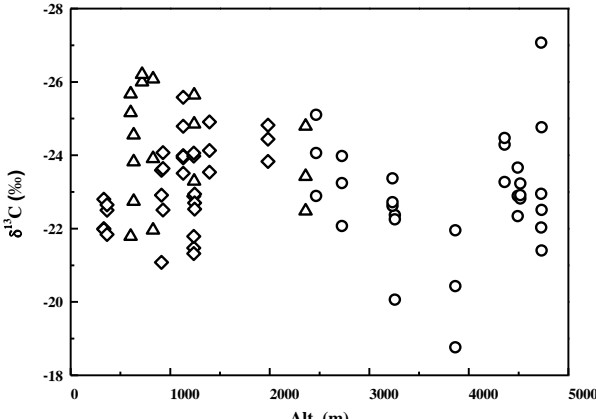

Fig.3 shows the variance in surface soil $\delta^{13}$C with altitude.







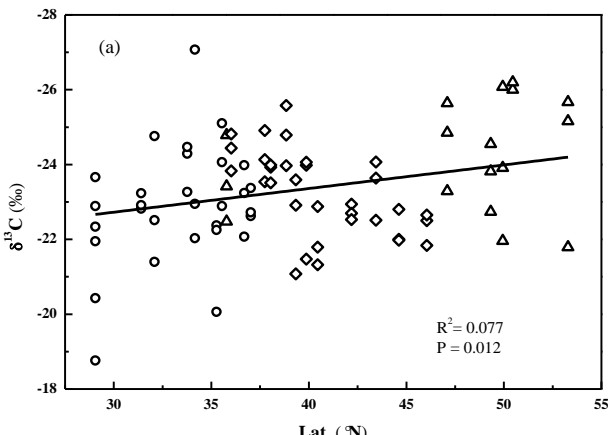


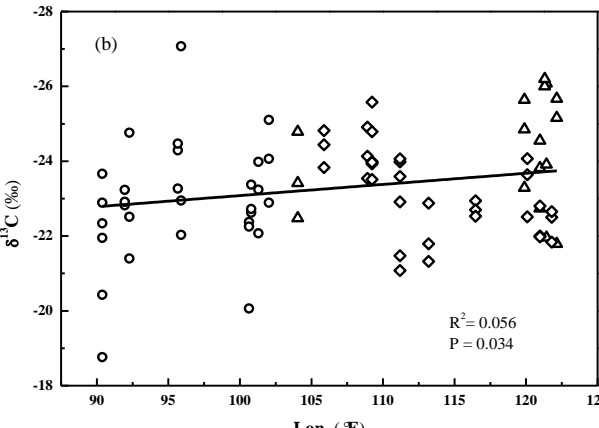


Fig.4 shows the relationships between the soil $\delta^{13}$C and latitude (a) and longitude (b).






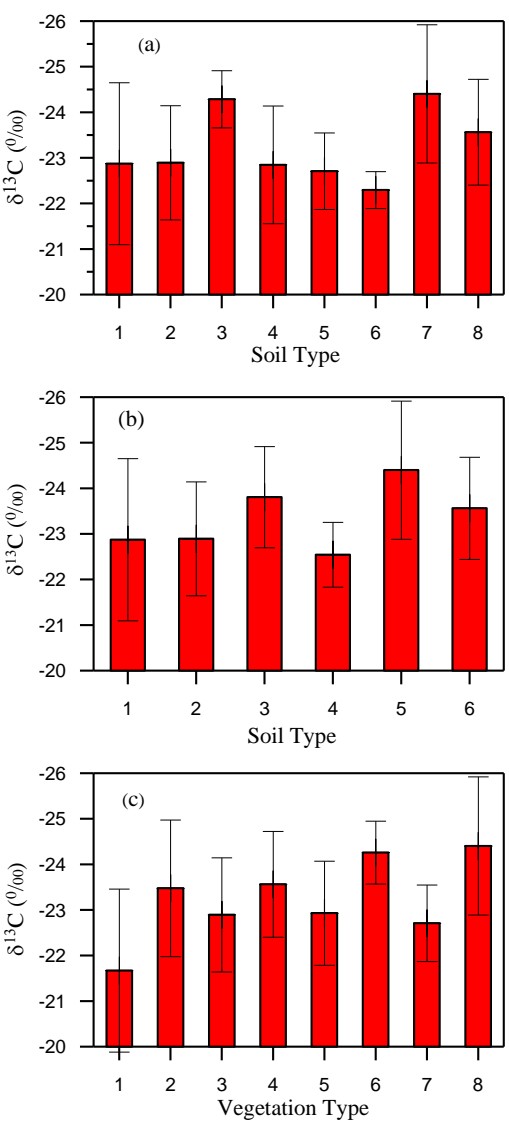


Fig.5 shows the effects of soil types and vegetation types on the soil $\delta^{13}$C. (a.) soil types based on
Chinese soil taxonomy. 1. Matti-Gelic Cambosols; 2. Hapli- Cryic Aridosolsl; 3. Calci-Orthic
Aridosols; 4. Mottlic Calci-Orthic Aridosols; 5. Typic Calci-Ustic Isohumosols; 6. Pachi-Ustic
Isohumosols; 7. Umbri-Gelic Cambosols; 8. Hapli-Ustic Argosols. (b.) soil types based on WRB.
1. Cambisols; 2. Leptosols; 3. Calcisols; 4. Chernozems; 5. Umbrisols; 6. Luvisols. (c.) vegetation
type. 1. Alpine grassland; 2. Alpine meadow; 3. Subalpine grassland; 4. Temperate coniferous and
broad-leaved mixed forests; 5. Temperate meadow steppe; 6. Semi-desert grasslands; 7. Temperate
typical steppe; 8. Frigid temperate coniferous forest. The bar in Fig.5 indicates ±1SD.





Table 1 Information of the sampling sites

| No. | Site name | MAT/°C | SMT/°C | MAP/mm | Alt./m | Lat./N° | Lon./E° | Mean δ13C (‰) | Vegetation type | Dominate species | Soil types |
|---|---|---|---|---|---|---|---|---|---|---|---|
| 1 | LangKaZi | -5.3 | 10.8 | 376 | 4492 | 29.06 | 90.39 | -23.0 | Alpine grassland | *Stipa*、*Festuca* and *Carex* | Matti-Gelic Cambosols (Cambisols) |
| 2 | RiKaZe | 6.3 | 14 | 420 | 3865 | 29.33 | 88.98 | -20.4 | Alpine grassland | *Stipa*、*Festuca* and *Carex* | Matti-Gelic Cambosols (Cambisols) |
| 3 | NaQu | -2.2 | 8.2 | 406 | 4519 | 31.41 | 91.96 | -23.0 | Alpine meadow | *Kobresia* | Matti-Gelic Cambosols (Cambisols) |
| 4 | NieRong | -1.2 | 6.3 | 400 | 4731 | 32.09 | 92.27 | -22.9 | Alpine meadow | *Kobresia* | Matti-Gelic Cambosols (Cambisols) |
| 5 | ZhiDuo | -0.5 | 7 | 394 | 4360 | 33.77 | 95.66 | -24.0 | Alpine meadow | *Kobresia* | Matti-Gelic Cambosols (Cambisols) |
| 6 | QuMaLai | -3.3 | 7.4 | 391.7 | 4727 | 34.16 | 95.9 | -24.0 | Alpine meadow | *Kobresia* | Matti-Gelic Cambosols (Cambisols) |
| 7 | TongDe | 1.2 | 15.8 | 371 | 3258 | 35.27 | 100.64 | -21.6 | Subalpine grassland | *Stipa* and *Hippolytia* | Hapli- Cryic Aridosolsl (Leptosols) |
| 8 | TongRen | 5.2 | 19.5 | 425.7 | 2467 | 35.55 | 102.03 | -24.0 | Subalpine grassland | *Stipa* and *Hippolytia* | Hapli- Cryic Aridosolsl (Leptosols) |
| 9 | HuangYuan | 3 | 13.9 | 408.9 | 2725 | 37.02 | 100.8 | -22.9 | Subalpine grassland | *Stipa* and *Hippolytia* | Hapli- Cryic Aridosolsl (Leptosols) |
| 10 | HaiYan | 1.5 | 9.5 | 400 | 3233 | 36.69 | 101.3 | -23.1 | Subalpine grassland | *Stipa* and *Hippolytia* | Hapli- Cryic Aridosolsl (Leptosols) |
| 11 | YuZhong | 6.6 | 20.1 | 403 | 2361 | 35.78 | 104.05 | -23.6 | Temperate coniferous and broad-leaved mixed forests | *Pinus tabulaeformis* | Hapli-Ustic Argosols (Luvisols) |
| 12 | XiJi | 5.3 | 16.8 | 400 | 1982 | 36.02 | 105.88 | -24.4 | Temperate meadow steppe | *Stipa* and *Hippolytia* | Calci-Orthic Aridosols(Calcisols) |
| 13 | JingBian | 7.8 | 20 | 395.4 | 1394 | 37.74 | 108.91 | -24.2 | Semi-desert grasslands | *Stipa*、*Hippolytia* and *Ajania* | Calci-Orthic Aridosols(Calcisols) |
| 14 | HengShan | 8.5 | 22.2 | 397 | 1131 | 38.04 | 109.24 | -23.8 | Semi-desert grasslands | *Stipa*、*Hippolytia* and *Ajania* | Calci-Orthic Aridosols(Calcisols) |
| 15 | ShenMu | 8.9 | 21.5 | 393 | 1131 | 38.84 | 110.44 | -24.8 | Semi-desert grasslands | *Stipa*、*Hippolytia* and *Ajania* | Calci-Orthic Aridosols(Calcisols) |
| 16 | HeQu | 8.8 | 22.3 | 426 | 912 | 39.33 | 111.19 | -22.5 | Temperate meadow steppe | *Bothriochloa* and *Pennisetum* | Mottic Calci-Orthic Aridosols(Calcisols) |
| 17 | ZhunGeErQi | 7.5 | 20 | 400 | 1236 | 39.87 | 111.18 | -23.2 | Temperate meadow steppe | *Stipa* and *Aneuralepidium* | Mottic Calci-Orthic Aridosols(Calcisols) |
| 18 | FengZhen | 4.7 | 19.4 | 413 | 1236 | 40.45 | 113.19 | -22.0 | Temperate typical steppe | *Stipa* and *Aneuralepidium* | Typic Calci-Ustic Isohumosols (Chernozems) |
| 19 | DuoLun | 2.4 | 17.7 | 407 | 1245 | 42.18 | 116.47 | -22.7 | Temperate typical steppe | *Stipa* and *Aneuralepidium* | Typic Calci-Ustic Isohumosols (Chernozems) |
| 20 | LinXi | 2.2 | 19.9 | 370 | 928 | 43.44 | 110.08 | -23.4 | Temperate typical steppe | *Stipa* and *Aneuralepidium* | Typic Calci-Ustic Isohumosols (Chernozems) |
| 21 | ZhaLuTeQi | 2.8 | 22.4 | 387 | 332 | 44.61 | 120.97 | -22.3 | Temperate meadow steppe | *Stipa*、*Aneuralepidium* and | Pachi-Ustic Isohumosols (Chernozems) |





| No. | Site | MAT | SMT | MAP | Alt | Lat | Lon | | Vegetation type | Dominant species | Soil type |
|---|---|---|---|---|---|---|---|---|---|---|---|
| 22 | WuLanHaoTe | 4.1 | 20.8 | 416 | 366 | 46.05 | 121.79 | -22.3 | Temperate meadow steppe | *Stipa , Aneuralepidium and Filifolium* | Pachi-Ustic Isohumosols (Chernozems) |
| 23 | AErShan | -2.7 | 15 | 391 | 1240 | 47.1 | 119.89 | -24.6 | Frigid temperate coniferous forest | *Larix gmelinii and Betula platyphylla Suk* | Umbri-Gelic Cambosols (Umbrisols) |
| 24 | YaKeShi | -2.9 | 18.5 | 379 | 634 | 49.33 | 120.97 | -23.7 | Frigid temperate coniferous forest | *Larix gmelinii* | Umbri-Gelic Cambosols (Umbrisols) |
| 25 | KuDuEr | -5.5 | 17.3 | 402 | 829 | 49.94 | 121.43 | -24.0 | Frigid temperate coniferous forest | *Larix gmelinii and Betula platyphylla Suk* | Umbri-Gelic Cambosols (Umbrisols) |
| 26 | GenHe | -5.3 | 17.9 | 424 | 718 | 50.46 | 121.31 | -26.1 | Frigid temperate coniferous forest | *Betula platyphylla Suk* | Umbri-Gelic Cambosols (Umbrisols) |
| 27 | BeiJicun | -5.5 | 18.4 | 450.8 | 603 | 53.29 | 122.15 | -24.2 | Frigid temperate coniferous forest | *Larix gmelinii and Pinus sylvestris var* | Umbri-Gelic Cambosols (Umbrisols) |

Note: MAT, SMT, MAP, Alt, Lat and Lon are the abbreviations of mean annual temperature, summer mean temperature, mean annual precipitation, altitude, latitude, longitude, respectively. Longitude, latitude and altitude of each site were from the portable GPS; MAT and MAP represent the average values of more than 30 years, SMT presents the average value of June, July and August for more than 30 years. All climatic data were from the local meteorological stations and the China Meteorological Data Sharing Service System (http://cdc.cma.gov.cn/shishi/climate.jsp); The soil types are based on Chinese soil taxonomy and WRB (in the brackets).






Table 2 shows the results from multiple regressions.

| Model | $R^2$ | Adjusted $R^2$ | F | p-value |
|---|---|---|---|---|
| 1 | 0.091 | 0.030 | 1.484 | 0.205 |
| 2 | 0.374 | 0.273 | 3.690 | < 0.001 |
| 3 | 0.297 | 0.195 | 2.911 | 0.004 |
| 4 | 0.362 | 0.247 | 3.164 | 0.001 |

Note: Model-1 is the multiple regression of soil $\delta^{13}C$ against MAT, MAP, altitude, latitude and longitude; For Model-2, Model-3 and Model-4, in addition to taking these five environmental factors as independent variables, the soil types based on Chinese nomenclature and WRB, and the vegetation types as dummy variables were separately introduced in the multiple regressions.
