# Peer review of "Temperature exerted no influence on the soil organic matter"

_Biogeosciences, 2015_

## Referee Comment (RC1) · M. Hodson (Referee) · 2 May 2016

Comments from Dr Martin J. Hodson:

General Comments: This work looks at the effect of temperature on soil $\delta$13C along the isopleth of 400 mm mean annual precipitation in China. It is a question that needs addressing. The methodology is appropriate and is clear. The research appears to have been very carefully conducted, and the authors clearly show that there was no effect of temperature, a negative result. Personally, I am in favour of publishing negative results, and actually some other factors, vegetation and soil type, did affect soil $\delta$13C so it is not all negative. It is a good paper, but I feel it needs some shortening of the discussion, and the English needs attention.

[Figure]

Specific Comments: Do we really need the paragraph beginning on line 285? If all the samples were taken along the isopleth of 400 mm mean annual precipitation we would expect rainfall to have little effect!! The whole discussion is a bit long and could be trimmed to make it more readable.

Technical Corrections: The standard of the English needs to be improved before final publication. Too many sentences are over long and confusing. I started to do some corrections, but there are too many, and this is not the job of a reviewer. I got to line 86:

Even the title is not quite right: Temperature exerted no influence on the organic carbon isotope of surface soil along the isopleth of 400 mm mean annual precipitation in China. Maybe this might be better as: Temperature exerted no influence on the soil organic matter $\delta13C$ of surface soil along the isopleth of 400 mm mean annual precipitation in China.

Line 34: soil $\delta13C$, and this

Line 42: Multiple regressions in which the above

Line 65: reconstructed high-resolution

Line 70: to our knowledge, no researchers

Line 73: While this may be likely

Line 75: delete "furthermore, these studies do not appear to result in a conclusive statement."

Line 83: $\delta13CSOM$ is generally close to plant $\delta13C$ despite

Line 86: Thus, plant $\delta13C$ might also influence $\delta13CSOM$. $\delta13C$ in plants, especially C3 plants, is tightly associated with precipitation and so precipitation may have an influence on soil $\delta13C$ (Diefendorf et al., 2010; Kohn, 2010).

---

## Author Comment (AC1) · 31 May 2016

Dear Dr. Martin J. Hodson, Many thanks for your comments. We have modified the manuscript following the comments.

Response to the comments: 1) The reviewer felt it needs some shortening of the discussion. We think the suggestion is great, thus, some unnecessary contents have been deleted in the new discussion. 2) The reviewer felt that the English needs attention. Thank you. We have asked Professor Eric Posmentier in department of Earth Sciences, Dartmouth College to do the English editing again, and he had edited the English thoroughly. Thus, I believe the English must be greatly improved. Actually, Professor Eric Posmentier has done English-editing of the manuscript last year, however,

we added some new contents in the manuscript after his editing, this may introduce some English mistakes. 3) The title was changed following the comment.

Best wishes

Sincerely yours,

Guoan Wang

Please also note the supplement to this comment:
http://www.biogeosciences-discuss.net/bg-2015-624/bg-2015-624-AC1-supplement.pdf

―――――――――――――――――

**Supplement:**

[revised manuscript text omitted]

Table 2 shows the results from multiple regressions.

| Model | $R^2$ | Adjusted $R^2$ | F | p-value |
|-------|-------|----------------|-------|---------|
| 1 | 0.091 | 0.030 | 1.484 | 0.205 |
| 2 | 0.374 | 0.273 | 3.690 | <0.001 |
| 3 | 0.297 | 0.195 | 2.911 | 0.004 |
| 4 | 0.362 | 0.247 | 3.164 | 0.001 |

Note: Model-1 is the multiple regression of soil $\delta^{13}C$ against MAT, MAP, altitude, latitude and longitude; For Model-2, Model-3 and Model-4, in addition to taking these five environmental factors as independent variables, the soil types based on Chinese nomenclature and WRB, and the vegetation types as dummy variables were separately introduced in the multiple regressions.

---

## Short Comment (SC1) · 11 Jun 2016

General Comments: This study presents d13C data of TOC from surface soil across a great temperature gradient along the 400-mm precipitation isoline in China, with the objective to assess the effect of temperature on soil d13C values. It clearly shows no significant influence of temperature on soil d13C. This finding is very important for the study of quantitative reconstruction of palaeoclimate using d13C of soil organic matter and therefore would be of great interest to the palaeoclimate and palaeoecology communities. This work appears to have been very carefully conducted, and it is a good paper, however, I think the discussion needs some shortening.

Specific Comments: Since all the soil samples were collected along the isopleth of 400

mm mean annual precipitation, no effect of precipitation would be expected, I feel it is not necessary to discuss the influence of precipitation, and suggest the discussion would be reduced or even removed.

Technical Corrections: Although the written English is generally understandable, there are some inappropriate expressions and typo errors that need to be improved and corrected before final publication.

---

## Author Comment (AC2) · 20 Jun 2016

A point-by-point response to Dr. SL Yang Dear Dr. SL Yang, Many thanks for your comments. We have modified the manuscript following the comments.

Response to the comments: 1) Your suggestion that the discussion needs some shortening is great, thus, some unnecessary contents about C4 plant distribution in China (in the first paragraph in the old version) have been deleted in the new discussion. In addition, we have deleted the discussion about the influence of precipitation on soil isotope. 2) The reviewer felt that the English needs some corrections. We had asked Professor Eric Posmentier in department of Earth Sciences, Dartmouth College to do the English editing again. We believe English was greatly improved in the new version.

[Figure]

Thank you.

Best wishes

Sincerely yours,

Guoan Wang

---

## Referee Comment (RC2) · Anonymous Referee #3 · 1 Jul 2016

Overall, I believe the conclusion of the paper to be strong. The author's found no effect of MAT on C13 isotopes across this latitude. While this confirms prior results, I think this is an important paper in the field and will affect the interpretation of carbon isotope results for both biologists and geologists. Even though the result is negative, it is an important negative result. The study was carried out carefully and the statistical analysis was appropriate.

That being said, the discussion wanders quite a bit and discusses several topics that are irrelevant to the paper or are obvious, such as the humidity cline and the plant variation. The discussion should focus mainly on the temperature and soil main effect and that would tighten it up and strengthen it.

[Figure]

Overall, there are several grammatical errors including missed commas, etc. The writing is okay, but could be improved and it needs to be retitled.

—————————————————

---

## Author Response (AR1)

Dear editor,

Many thanks. We have already improved the manuscript according to the reviewer comments and the interactive discussion suggestions. Also we have improved English by Editing Service company.    Thus, we sincerely hope to publish it in BG journal.

Best Regards,

Guoan Wang

**A point-by-point response Referee #1**

Dear Dr. Martin J. Hodson,
Many thanks for your comments. We have modified the manuscript following the comments.

Response to the comments:
1) The reviewer felt it needs some shortening of the discussion. We think the suggestion is great, thus, some unnecessary contents have been deleted in the new discussion.

2) The reviewer felt that the English needs attention. Thank you. We have asked Professor Eric Posmentier in department of Earth Sciences, Dartmouth College to do the English editing again, and he had edited the English thoroughly. Thus, I believe the English must be greatly improved. Actually, Professor Eric Posmentier has done English-editing of the manuscript last year, however, we added some new contents in the manuscript after his editing, this may introduce some English mistakes.

3) The title was changed following the comment.

Best wishes

Sincerely yours,

Guoan Wang

**A point-by-point response to Referee #2**

Dear Dr. SL Yang,

Many thanks for your comments. We have modified the manuscript following the
comments.

Response to the comments:

1)  Your suggestion that the discussion needs some shortening is great, thus, some
unnecessary contents about $C_4$ plant distribution in China (in the first paragraph in
the old version) have been deleted in the new discussion. In addition, we have
deleted the discussion about the influence of precipitation on soil isotope.

2)  The reviewer felt that the English needs some corrections. We had asked Professor
Eric Posmentier in department of Earth Sciences, Dartmouth College to do the
English editing again. We believe English was greatly improved in the new
version. Thank you.

Best wishes

Sincerely yours,

Guoan Wang

## A point-by-point response to Referee #3

Dear Referee #3.

Many thanks for your comments. We have modified the manuscript following the
comments.

**Comment-1**: That being said, said that the discussion wanders quite a bit and discusses
several topics that are irrelevant to the paper or are obvious, such as the humidity cline
and the plant variation. The discussion should focus mainly on the temperature and soil
main effect and that would tighten it up and strengthen it.

**Response**: We think the suggestion is great, thus, the influence of precipitation was
deleted in new discussion (please see the lines 297-307 in the text with a mark-up);
the content with respect to plant variation was greatly shortened (please see the lines
244-247 and 251-256 in the text with a mark-up).

**Comment-2**: Overall, there are several grammatical errors including missed commas,
etc. The writing is okay, but could be improved and it needs to be retitled.

**Response**: Thank you. In order to improve the English, we asked an English service
company, Editage Company, to edit the manuscript thoroughly. Referee #3 suggested that it needs to be retitled. Referee #1 also suggested us to change the title. Thus, we
modified the title following their suggestions.

Best wishes

Sincerely yours,

[revised manuscript text omitted]

Note: MAT, SMT, MAP, Alt, Lat and Lon are the abbreviations of mean annual temperature, summer mean temperature, mean annual precipitation, altitude, latitude, longitude, respectively. Longitude, latitudeand altitude of each site were from the portable GPS; MAT and MAP represent the average values of more than 30 years,

SMT presents the average value of June, July and August for more than 30 years. All climatic data were from the local meteorological stations and the China

Meteorological Data Sharing Service System (http://cdc.cma.gov.cn/shishi/climate.jsp); The soil types are based on Chinese soil taxonomyand WRB (in the brackets).

Table 2 shows the results from multiple regressions.

| Model | $R^2$ | Adjusted $R^2$ | F | p-value |
|---|---|---|---|---|
| 1 | 0.091 | 0.030 | 1.484 | 0.205 |
| 2 | 0.374 | 0.273 | 3.690 | <0.001 |
| 3 | 0.297 | 0.195 | 2.911 | 0.004 |
| 4 | 0.362 | 0.247 | 3.164 | 0.001 |

Note: Model-1 is the multiple regression of soil $\delta^{13}C$ against MAT, MAP, altitude, latitude and longitude; For Model-2, Model-3 and Model-4, in
addition to taking these five environmental factors as independent variables, the soil types based on Chinese nomenclature and WRB, and the
vegetation types as dummy variables were separately introduced in the multiple regressions.

---

## Author Response (AR2)

Dear editor,

Many thanks for your comments. We have asked the Editage Editing Service Company to edit the English thoroughly once again (please see the following letter from the editor of this company). Meantime, we have inserted figures into manuscript. Thus, we sincerely hope to publish it in BG journal. Please contact me freely if you have any questions.

Best Regards,

Guoan Wang

[Figure]

Letter from the editor

**Message from your editor, Thomas**

Dear Author,

It was a pleasure working on your document. Do go through my changes and comments in the edited file, as well as the notes in this document.

Please send me your feedback or any questions through your Editage Online account (online.editage.cn).

**A point-by-point response Referee #1**

Dear Dr. Martin J. Hodson,
Many thanks for your comments. We have modified the manuscript following the comments.

Response to the comments:
1) The reviewer felt it needs some shortening of the discussion. We think the suggestion is great, thus, some unnecessary contents have been deleted in the new discussion.

2) The reviewer felt that the English needs attention. Thank you. We have asked Professor Eric Posmentier in department of Earth Sciences, Dartmouth College to do the English editing again, and he had edited the English thoroughly. Thus, I believe the English must be greatly improved. Actually, Professor Eric Posmentier has done English-editing of the manuscript last year, however, we added some new contents in the manuscript after his editing, this may introduce some English mistakes.

3) The title was changed following the comment.

**A point-by-point response to Referee #2**

Dear Dr. SL Yang,

Many thanks for your comments. We have modified the manuscript following the comments.

Response to the comments:

1) Your suggestion that the discussion needs some shortening is great, thus, some unnecessary contents about $C_4$ plant distribution in China (in the first paragraph in the old version) have been deleted in the new discussion. In addition, we have deleted the discussion about the influence of precipitation on soil isotope.

2) The reviewer felt that the English needs some corrections. We had asked Professor Eric Posmentier in department of Earth Sciences, Dartmouth College to do the English editing again. We believe English was greatly improved in the new version. Thank you.

**A point-by-point response to Referee #3**

Dear Referee #3.

Many thanks for your comments. We have modified the manuscript following the comments.

**Comment-1**: That being said, said that the discussion wanders quite a bit and discusses several topics that are irrelevant to the paper or are obvious, such as the humidity cline and the plant variation. The discussion should focus mainly on the temperature and soil main effect and that would tighten it up and strengthen it.

**Response**: We think the suggestion is great, thus, the influence of precipitation was deleted in new discussion; the content with respect to plant variation was greatly shortened.

**Comment-2**: Overall, there are several grammatical errors including missed commas, etc. The writing is okay, but could be improved and it needs to be retitled.

**Response**: Thank you. In order to improve the English, we asked an English service company, Editage Company, to edit the manuscript thoroughly. Referee #3 suggested that it needs to be retitled. Referee #1 also suggested us to change the title. Thus, we modified the title following their suggestions.

Best wishes
Sincerely yours,
Guoan Wang

[revised manuscript text omitted]